



# AntAir: satellite-derived 1 km daily Antarctic air temperatures since 2003

Hanna Meyer[1], Marwan Katurji[2], Florian Detsch[3], Fraser Morgan[4], Thomas Nauss[5], Pierre Roudier[6,7], and Peyman Zawar-Reza[2]

[1]Institute of Landscape Ecology, Westfälische Wilhelms-Universität Münster, 48149 Münster, Germany
[2]Center for Atmospheric Research, University of Canterbury, Christchurch 8020, New Zealand
[3]METER Group AG, 81379 München, Germany
[4]Environmental Informatics, Landcare Research, Auckland 1142, New Zealand
[5]Environmental Informatics, Faculty of Geography, Philipps-Universität Marburg, 35037 Marburg, Germany
[6]Soils & Landscapes, Landcare Research, Palmerston North 4442, New Zealand
[7]Te Punaha Matatini, a New Zealand Centre for Research Excellence, Private Bag 92019, Auckland 1142, New Zealand

*Correspondence to:* Hanna Meyer (hanna.meyer@uni-muenster.de)

**Abstract.** Air temperature is an important baseline parameter for terrestrial Antarctica in the context of patterns and processes in climatology, hydrology or ecology. There are still large uncertainties on how the Antarctic system responds to spatio-temporal variability of temperature. This can partly be attributed to the lack of high resolution datasets. In this paper, we present AntAir, a new dataset of gridded air temperatures in 1 km spatial and daily temporal resolution that is available since 2003. AntAir was created by modelling daily air temperature from MODIS land surface temperature using machine learning algorithms. Data from 70 weather stations was used as a reference. Daily temperatures could be estimated with a $R^2$ of 0.91 and a RMSE of 5.07°C validated on independent years. The performance to estimate the time series of a new spatial location was $R^2$=0.78 and RMSE=5.83°C. Hence the high spatial and temporal resolution of the dataset as well as the high accuracy make AntAir an important baseline dataset for a wide range of applications in environmental science of Antarctica. The dataset is available at https://doi.pangaea.de/10.1594/PANGAEA.902166 (daily, Meyer et al., 2019a) and https://doi.pangaea.de/10.1594/PANGAEA.902193 (monthly, Meyer et al., 2019b)

## 1 Introduction

Near-surface air temperature in Antarctica is an important driver of terrestrial biodiversity (Convey, 2010) and is decisive for hydrological (Herbei et al., 2016) and glaciological processes (Cook et al., 2005). Under a warming global climate Antarctica has been the focus of climate scientists for its impacted melting ice shelves, surface mass balance, sea ice changes (Siegert, 2016; Favier et al., 2017; Steig et al., 2013, 2009), and changes in atmospheric circulation patterns promoting higher meridional transport of atmospheric moisture and heat towards the Antarctic coastline (Turner et al., 2013; Raphael et al., 2016; Marshall et al., 2017). In Antarctica, there is also growing evidence of localized climate extremes caused by foehn wind warming events (Speirs et al., 2012; Zawar-Reza et al., 2013) that are also impacting hydrological extremes in areas of high biodiversity (Doran et al., 2008) and ice shelf mass balance (Cape et al., 2015; Elvidge et al., 2015) and smaller scale temperature variability due to





surface type, cover, or topographic temperature moderation like cold air pooling or localized thermally driven winds (Katurji et al., 2013; Zawar-Reza and Katurji, 2014).

Extreme inter- and intra-annual meteorological events, shorter term climate perturbations, and continental scale climate variability can have an impact on habitat change and species extinction (Convey et al., 2018) and impact species that explore regional and continental scale breeding and feeding patterns (Cimino et al., 2016). In addition to understanding biodiversity, high resolution air temperature data are needed to support better ecosystem resilience understanding and biological conservation planning, policy and management (Terauds and Lee, 2016), and by designating more evidence based and newly proposed Antarctic protected areas (Shaw et al., 2014).

Despite the undoubted importance of air temperature for many ecosystem processes, there are still large uncertainties on how the Antarctic physical and biological system responds to spatio-temporal variability. This can partly be attributed to the lack of high resolution datasets on air temperature, thus research effort for its provision is in high demand (Schneider and Reusch, 2016).

Weather stations provide sub-daily measurements, however, their low spatial density does not allow for comprehensive spatio-temporal analysis. While climate models can be used to fill these gaps, their coarse spatial resolution is not capable of representing the important processes causing near-surface climate variability. Topography perturbs the synoptic flow patterns and cause localized extreme temperatures modulated by sub-regional and microscale processes. Remote sensing data and methods are therefore inevitable as they provide fine resolution and spatially explicit proxies for air temperature suitable to assess such small-scale dynamics. The MODerate-resolution Imaging Spectroradiometer (MODIS) onboard the two satellites Terra and Aqua provides suitable data for polar regions as it provides images for entire Antarctica several times of the day and with a spatial resolution of 1km. The MODIS Land Surface Temperature product (LST) has shown its suitability as a proxy for air temperature in various regions of the world (Colombi et al., 2007; Neteler, 2010; Benali et al., 2012; Emamifar et al., 2013; Hengl et al., 2012; Kilibarda et al., 2014; Kitsara et al., 2018). Hooker et al. (2018) recently developed a global gridded air temperature dataset from MODIS LST, however Antarctica is not included and the data have a monthly temporal resolution which is too coarse to observe certain environmental processes.

The "AntAir" dataset presented in this study uses the MODIS LST dataset for Antarctica and is based on the findings from a pilot study of Meyer et al. (2016) to provide spatially explicit and high resolution time series of near surface air temperature of terrestrial Antarctica. MODIS LST as well as terrain and solar properties are used as predictor variables for air temperature. Using machine learning, the nonlinear relationships between these predictors and daily air temperature measured by 70 weather stations between 2003 and 2017 are modelled and used to make spatio-temporal predictions of air temperature. The resulting dataset covers the entire Antarctica at a spatial resolution of 1 km and is available since 2003 (currently processed until 2016). It therefore represents a baseline for the investigation of local scale near-surface air temperature variability at a spatial and temporal resolution that allows detecting scale processes that are otherwise not well represented in current regional and global climate models. As air temperature is the driver for many ecosystem processes, the dataset can be regarded as a baseline data for various kinds of studies aiming at the detection or explanation of patterns and trends in climatological, ecological or hydrological processes.



## 2 Methods

AntAir was created based on MODIS and auxiliary data that were related to measured air temperature using machine learning algorithms. The steps towards the final AntAir dataset comprised the compilation and preprocessing of the weather station records as well as of the satellite-based predictor variables, variable selection, model training and the final model prediction on each daily MODIS dataset to create daily gridded air temperature data.

### 2.1 Data and preprocessing

The following section describes the datasets being used to create AntAir as well as the preprocessing steps required in view to the subsequent machine learning based modelling procedure.

#### 2.1.1 Weather station data

Four sources of automatic weather station data were used as reference data in this study. The Antarctic Meteorological Research Center (AMRC) at the University of Wisconsin (Lazzara et al., 2012) provides data from weather stations distributed over the entire continent. Further data from the McMurdo Dry Valleys were obtained from the Long Term Ecological Research (LTER) programme (Doran et al., 1995). Data from the Victoria Land and in the Eastern sector of the Antarctic Plateau were received from the Antarctic Meteo-Climatological Observatory of Italy. Each of the above mentioned measured air temperature in 3 m above the ground. Further, the United States Department of Agriculture (USDA) provides data from weather stations in the Ross Sea Region (Seybold et al., 2009; Soil Survey Staff). Air temperature from the USDA sites was measured at 1.6 m above ground. Temperature sensors of all providers were mounted within radiation shields.

In total, 70 weather stations obtained from these four sources were used for model training and validation (Fig. 1). All weather stations provide data in 15 minutes or hourly temporal resolution and were for this study aggregated to daily averages.

#### 2.1.2 MODIS LST

The daily LST data (version 5) (Wan, 2008) based on the MODIS sensor onboard the Aqua and Terra satellites are distributed as the MOD11A1 and MYD11A1 products (Land Processes Distributed Active Archive Center (LP DAAC), 2013). The products consist of daytime and nighttime measurements at 1 km spatial resolution. Their calculation is based on a split-window algorithm that uses the emissivities from MODIS bands 31 and 32 that were, in turn, calculated using information about land cover type, atmospheric column water vapour and lower boundary air surface (Wan, 2008). The data are cloud-masked using the MODIS Cloud Mask algorithm (Ackerman et al., 1998) that applies typical thresholds in the visible and infrared channels.

In this study we use daily sensor integrated daytime and nighttime data as major predictor variables to estimate air temperature. Thus for each day, one MODIS LST daytime and one MODIS LST nighttime data layer was used that reflects the aggregated records of both Terra and Aqua. The integration of Terra and Aqua was done on a pixel basis: if both Aqua and Terra were cloud free for the respective pixel then the mean value of both records was adopted, if only one of them was cloud free, the cloud free pixel was adopted.



**Figure 1.** Locations of the weather stations used as reference in this study overlaid on a Landsat composite (U.S Geological Survey, 2007). The color indicates the source of the data either from the the Antarctic Meteo-Climatological Observatory of Italy, the Long Term Ecological Research (LTER) programme, the United States Department of Agriculture (USDA) or the Antarctic Meteorological Research Center at the University of Wisconsin (UWISC).



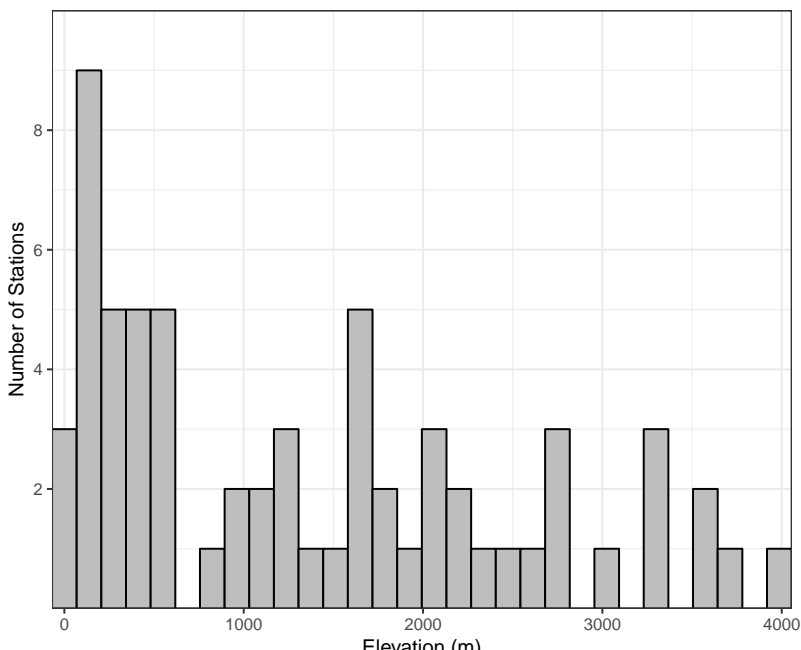

**Figure 2.** Histogram showing the gradient of elevation the weather stations are located on.

### 2.1.3 Elevation data, Solar information and hillshading properties

Elevation was included as further potential predictor variable for air temperature beside MODIS LST. The Radarsat Antarctic Mapping Project (RAMP) digital elevation model (DEM) (Liu et al., 2015), version 2, was used. The 200 m resolution DEM was bilinearly resampled to 1000 m to match the resolution of the MODIS LST data.

5    As proxy for solar radiation, the daily minimum, mean and maximum sun elevation was calculated on a pixel basis using the R implementation of Kelley and Richards (2018). Hillshading was calculated from slope and aspect of the terrain following the calculation of Hijmans (2017). Here, the sun elevation angle and azimuth on a pixel basis were applied to calculate the hillshading according to the respective solar information for each pixel.

Additionally, land cover information was included based on the Bedmap2 data (Fretwell et al., 2013) that were used to
10  classify the landscape into ice covered or ice free areas according to their ice surface elevation information.

### 2.2 Compilation of the training and test data set

MODIS LST data and the auxiliary predictors were extracted for the locations of the weather stations. The daily MODIS LST values were matched with the corresponding station records of air temperature. In total 142630 cloud free data points of matching daily MODIS data as well as measured air temperature derived from 70 weather stations and the period 2003 to the
15  first half of 2017 were derived. From these 15 years, every third year was used for model testing (2003, 2006, 2009, 2012,



2015), all other years were used for model training. The selected training years provided 92649 data points for model training. 49981 data points were available for model testing.

## 2.3 Machine learning based model training

AntAir uses a statistical model between measured air temperature and the spatio-temporal predictor variables from MODIS
and auxiliary data. The relative ability of different machine learning algorithms was tested to model the nonlinear and complex relationships between air temperature and the auxiliary predictors from past observations to make predictions of air temperature in space and time. The best model (based on spatial, temporal and visual performance assessment) was used for the creation of the AntAir dataset (Fig. 3).

### 2.3.1 Algorithms

Four different machine learning algorithms were tested for their suitability to predict air temperature: Random Forests (RF) (Breiman, 2001), Generalized Boosted Regression Models (GBM)(Friedman, 2002), Neural Networks (NNET) and Partial Least Squares Regression (PLS). These algorithms were chosen because they are either frequently applied algorithms (RF, NNET), or have shown good performance in the pilot study of Meyer et al. (2016) (GBM). PLS was further chosen to test a less complex algorithm. For these algorithms, the implementations of the R packages "randomForest" (Liaw and Wiener,
2002), "gbm" (Ridgeway, 2017), "nnet" (Venables and Ripley, 2002) and "pls" (Mevik et al., 2018) were used.

### 2.3.2 Feature selection

Meyer et al. (2018) have shown that spatio-temporal machine learning algorithms are sensitive to overfitting due to misinterpretations of certain predictors caused by spatio-temporal autocorrelation. To overcome this problem, predictor variables were analyzed for their relevance in view to spatio-temporal predictions by a forward feature selection in conjunction with
spatial cross-validation (CV) as implemented in the CAST package (Meyer, 2018). The algorithm first trains models (i.e. RF) of all possible 2-variable combinations of the total set of predictor variables. The best initial model in view to target-oriented performance is kept. The number of predictor variables is then iteratively increased. The improvement of the model is tested for each additional predictor using spatial CV. The process stops when none of the remaining variables decreases the error of the currently best model within one standard error of best model with less variables.
A 10-fold Leave-Location-Out (LLO) CV was used, following Meyer et al. (2018). Accordingly, the dataset was split into 10 folds. Each fold contained the complete time series of 1/10 of all weather stations. Models were then repeatedly trained by using the data of all except one fold and testing the model performance for the held-back data. Hence during feature selection, the optimal variables are identified as those that lead in combination to the lowest LLO CV error which means that they produce best results in predicting air temperature for new locations.



**Figure 3.** Overview of the modelling steps performed to create AntAir. The total data set represents 142630 cloud-free data points of daily MODIS LST data matched to the measured daily air temperatures at 70 weather stations in the years 2003 to 2017. The dataset also contains the further potential predictor variables such as elevation, solar properties and hillshading properties.



### 2.3.3 Final model training and model selection

After feature selection, the models were fine tuned to set up a final model for air temperature prediction. Using RF requires the number of predictor variables randomly selected at each split ("mtry") to be tuned (Kuhn and Johnson, 2013; James et al., 2013). While during computation time expensive feature selection mtry was only tuned for three different values distributed

between 2 and the number of predictor variables, the final model was extensively tuned. Therefore mtry was varied for each value between two and the number of predictor variables. GBM was tuned using an interaction depth between three and number of predictor variables and a number of trees between 100 and 500. Shrinkage was varied between 0.01 and 0.05. The number of neurons in the hidden layer of the NNET was tuned between two and the number of predictor variables and weight decay was varied between 0 and 0.1 with increment 0.01. For PLS, the number of components was tuned between 1 and the number

of predictor variables. See Kuhn and Johnson (2013) for further information on the hyperparameters.

The error of the final models was assessed using LLO CV. In contrast to the CV setup for feature selection, during final model training the number of folds was equal to the number of weather stations being used. Hence the LLO performance reflects how well the air temperature time series of a new location could be modelled.

The fitted models were used to create spatially explicit time series of air temperature. The models of the different algorithms

were then compared by statistical spatial and temporal performance as well as visually by inspecting the spatial predictions. The best performing model was then chosen for AntAir.

## 3   Results

The first requirement to the model is its ability to predict beyond the locations of the weather stations that were used for model training, since the model is used to generate predictions over the entire continent. The second requirement is that the

model must be able to predict beyond the range of years used for model training since the model should be applied beyond the training phase of 2017. Therefore, both spatial and temporal validation procedure were used. As the number of weather stations available for model setup is limited, the ability of the model to predict beyond the locations of weather stations was assessed by the spatial CV described above.

The ability of the model to predict beyond the points in time used for model training could be assessed using an external

validation since enough years were available to split into years used for model training and model validation. Therefore, the years that have not been randomly selected for model training (see above) were used for the external validation. The model was therefore applied to the years that have not been used for training and the predicted air temperature at the locations of the climate stations was compared to the measured air temperature. For CV as well as for the external validation root mean squared error (RMSE) and $R^2$ were used as major metrics describing the model performance.

A comparison between the algorithms revealed that RF and GBM were the best performing algorithms (Table 1, 2) with RF being superior in the temporal prediction.



**Table 1.** Spatial performance of the different algorithms Random Forests (RF), Generalized Boosted Regression Models (GBM), Neural networks (NNET) and Partial Least Squares Regression (PLS) estimated based on Leave-One-Station-Out cross-validation. The best performances are highlighted. The * indicates the performance of the GBM model, that was chosen for the creation of AntAir.

| Model | By Fold | | | Global | | |
|-------|---------|------|-------|--------|------|-------|
| | MAE | RMSE | $R^2$ | MAE | RMSE | $R^2$ |
| PLS | $4.58 \pm 1.62$ | $6.07 \pm \mathbf{1.93}$ | $0.77 \pm 0.16$ | 5.98 | 4.42 | 0.87 |
| RF | $\mathbf{4.38} \pm 1.70$ | $\mathbf{5.60} \pm \mathbf{1.93}$ | $\mathbf{0.81} \pm \mathbf{0.13}$ | 5.81 | 4.31 | 0.88 |
| GBM* | $\mathbf{4.38} \pm \mathbf{1.55}$ | $5.83 \pm 1.94$ | $0.78 \pm 0.16$ | **5.67** | **4.12** | **0.89** |
| NNET | $4.46 \pm 1.64$ | $5.87 \pm 1.86$ | $0.78 \pm 0.15$ | 5.83 | 4.32 | 0.88 |

**Table 2.** Temporal performance of the different algorithms Random Forests (RF), Generalized Boosted Regression Models (GBM), Neural networks (NNET) and Partial Least Squares Regression (PLS) estimated based on 5 years that have not been included in the model training. The best performances are highlighted. The * indicates the performance of the GBM model, that was chosen for the creation of AntAir.

| Model | MAE | RMSE | $R^2$ |
|-------|-----|------|-------|
| PLS | $4.33 \pm 0.02$ | $5.85 \pm 0.35$ | 0.88 |
| RF | $\mathbf{3.02} \pm \mathbf{0.01}$ | $\mathbf{4.24} \pm \mathbf{0.19}$ | **0.94** |
| GBM* | $3.69 \pm 0.02$ | $5.07 \pm 0.29$ | 0.91 |
| NNET | $4.04 \pm 0.02$ | $5.46 \pm 0.29$ | 0.90 |

### 3.1 Visual validation and final model justification

Statistically, RF was the best performing algorithm considering the temporal evaluation and together with GBM statistically strong in the spatial prediction. However, Appelhans et al. (2015) and Meyer et al. (2019c) have shown that a visual inspection is a further important validation step and should not be neglected in the selection of models. In a visual inspection of the
5  predictions, it was apparent that the RF predictions were strongly driven by linear bands caused by the hillshading and solar variables (Fig. 4). Especially the tree-based nature of RF caused abrupt transitions in air temperature which seem inappropriate considering the request of seamless data. Hence, despite RF being superior in the temporal prediction, the GBM based model was chosen as final model for the creation of AntAir. It featured smoother spatial transitions and still had a high statistical performance. It has also shown to be the most promising algorithm within the Pilot study of AntAir (Meyer et al., 2016).

10  ### 3.2 Detailed validation of the final model

The final GBM model is based (after spatial feature selection) on the predictor variables LST during daytime, LST during nighttime, daily maximum hillshade, daily mean sun elevation, daily maximum sun elevation and land cover (ice/no ice). Tuning revealed the optimal settings of 300 trees, an interaction depth of 13, and a shrinkage of 0.05. The ability of the model to predict air temperature of a new location could be described by an average $R^2$ of 0.78 and a RMSE of 5.83°C (Table 1).

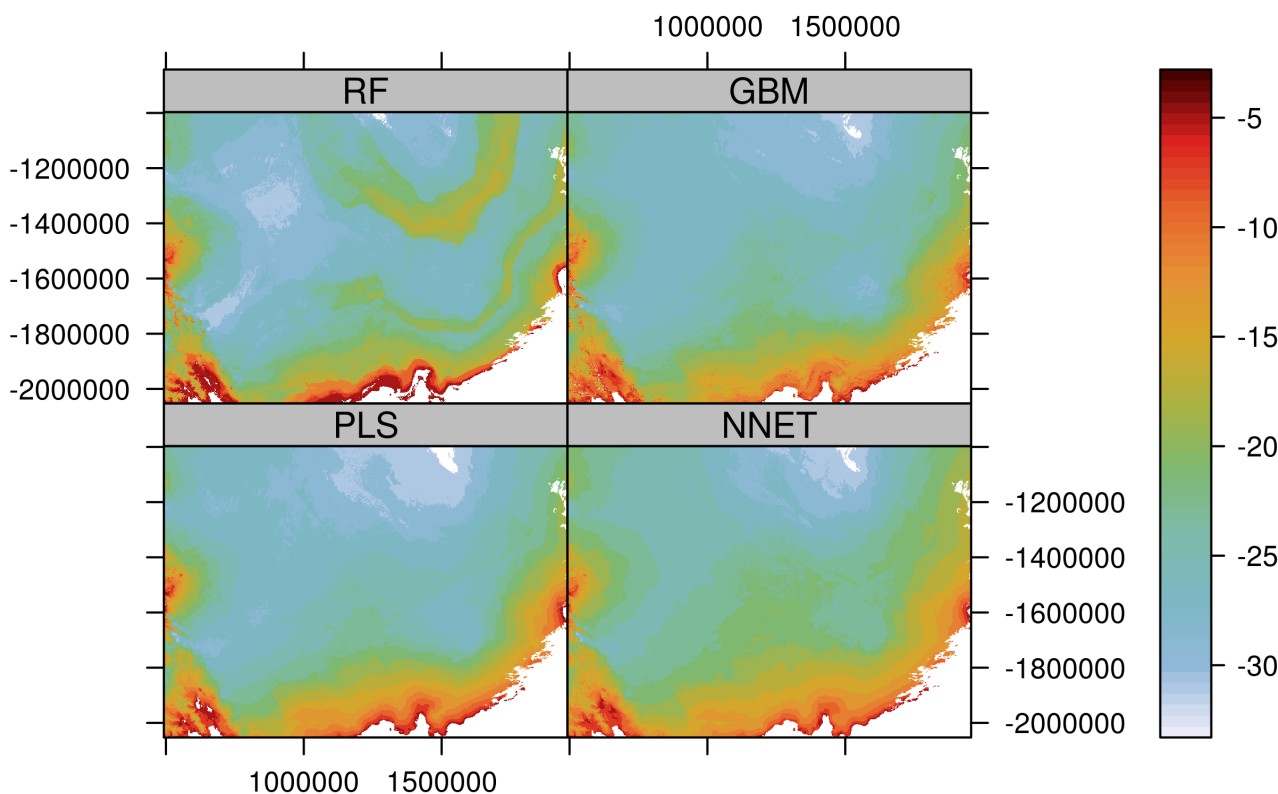

**Figure 4.** Visual comparison of the spatial air temperature predictions (in °C) made by the four machine learning algorithms Random Forests (RF), Generalized Boosted Regression Models (GBM), Partial least squares regression (PLS) and neural networks (NNET). The maps represents a small extent of the Antarctic region in which the artefacts in the RF predictions are most obvious.

Comparing all spatially independent predictions together, the $R^2$ was 0.89 and the RMSE 5.67°C. The external validation on new years indicated a performance of $R^2 = 0.91$ and RMSE = 5.07°C (Table 2). The air temperature gradient could be well modelled (Fig. 5).

When the daily modelled air temperature was aggregated to monthly and yearly means, the mean absolute error decreased from 3.69 °C (daily) to 2.24 °C (monthly) to 1.73 °C (yearly)(Fig. 6a).

The model was especially accurate during summer months (Fig. 6b) while the errors were higher in winter. This can be attributed to the higher impact of wind in winter which causes air temperature to change rapidly.

Fig. 7 shows the resulting long-term mean (2003-2016) for the entire extent of Antarctica, along with the frequently studied area of the McMurdo Dry Valleys, which highlights the spatial detail offered by the dataset.

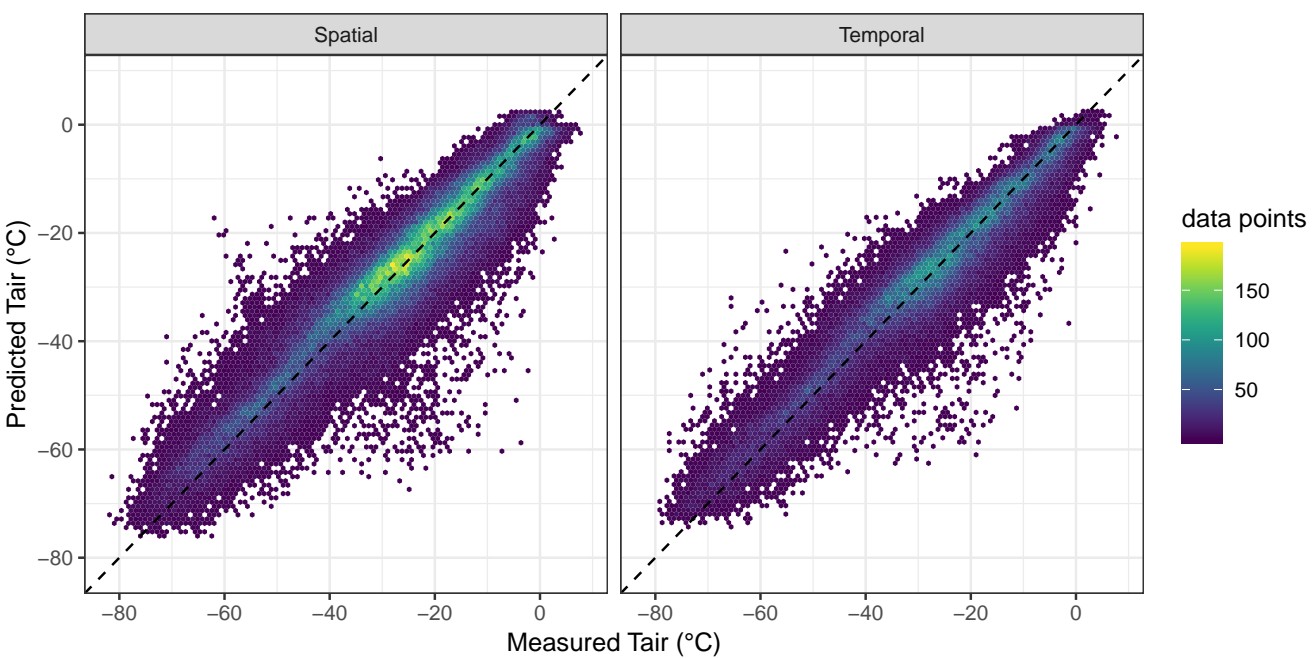

**Figure 5.** Comparison between measured and modelled air temperature based on (a) spatial cross validation and (b) external validation using all data of 5 years that have not been included in the model training.

## 4 Conclusions

The validation of the model underlines the high value of AntAir as a high resolution air temperature dataset for entire Antarctica. As air temperature is an important driver of ecosystem processes in Antarctica, the potential applications of AntAir are diverse. In a recent study, AntAir has been applied to detect foehn warming episodes within the McMurdo Dry Valleys (Katurji

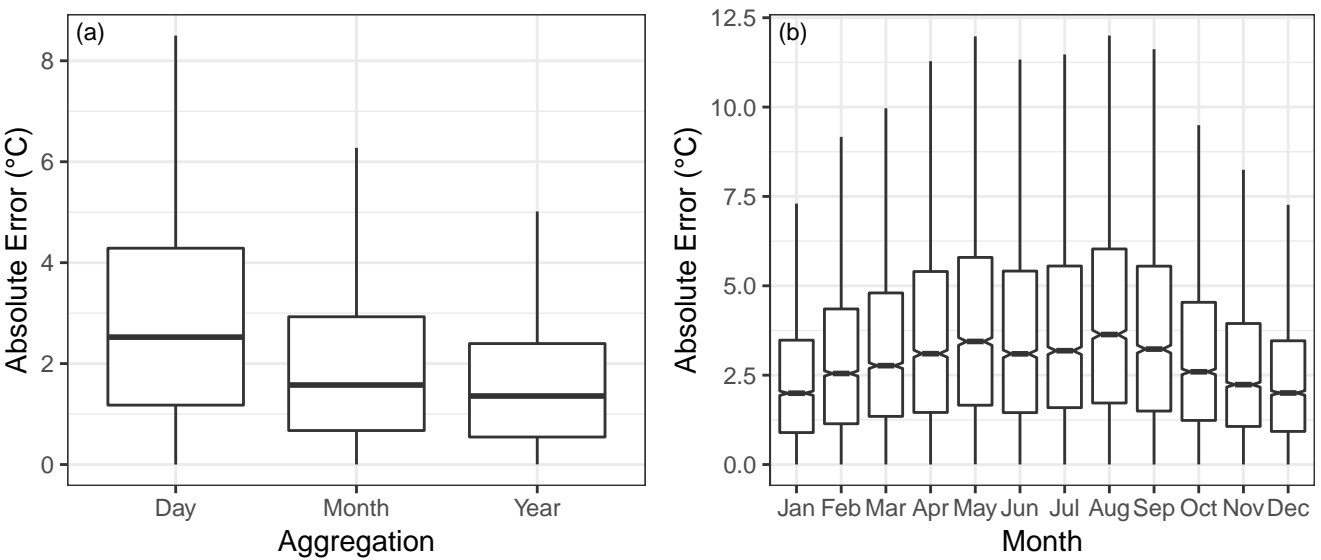

**Figure 6.** Absolute error of the modelled air temperature on a monthly or yearly aggregation level compared to the daily error (a) or grouped by the month of the year (b). Each box includes all daily/monthly/yearly temperatures of the 5 years used for model validation and each of the weather stations. Outliers are removed in this figure to highlight the general differences.

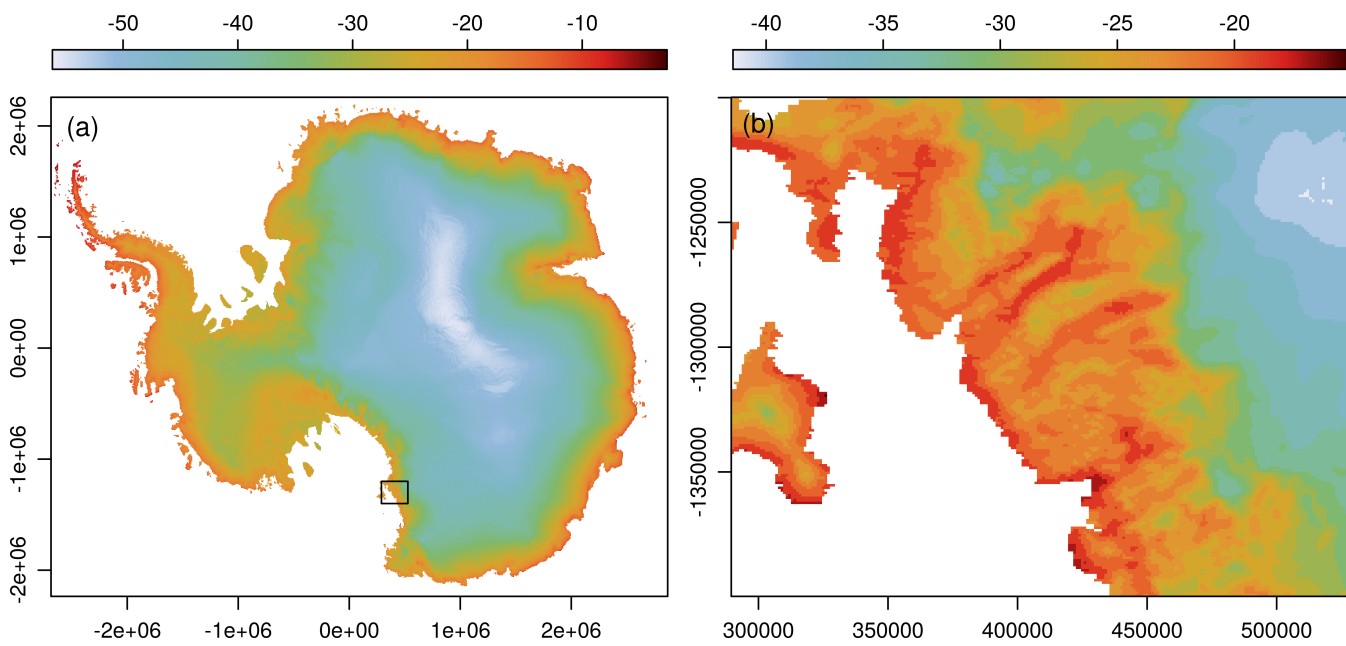

**Figure 7.** Long-term (2003-2016) average air temperature in°C for (a) entire Antarctica and (b) with focus on the McMurdo Dry Valleys. Rectangle in a shows the location of b.





et al., 2018). It will certainly be an important baseline product to detect recent trends in air temperature in the context of climate change but also is of important value for biologists in explaining diversity patterns.

Despite the good validation results, one limitation of AntAir is that no air temperature could be given for cloudy conditions due to the need of underlying MODIS data that are not available when land is covered by clouds. Future efforts on improvement
of the dataset will therefore focus on reliable spatio-temporal interpolation methods in order to provide seamless air temperature data in an envisaged next version of AntAir.

The validation further showed that the model was less accurate in winter months which is understandable considering the highly variable temperature conditions in winter caused by wind that could only partly be reflected by AntAir. Future directions towards a second version of the product will therefore be on the inclusion of climate model based wind properties
as an additional predictor which might support the model in highly dynamic winter conditions.

## 5   Code and data availability

The AntAir dataset is available from a PANGAEA repository (Meyer et al., 2019a). The dataset covers the entire continent of Antarctica in 1 km spatial resolution for the period 2003 to the first half of the year 2017. Each year consists of 365 raster layers that represent the predicted air temperature for the respective day. The raster layers are distributed using the GeoTIFF
format and the Antarctic Polar Stereographic projection (EPSG 3031). The unit of the values is degree Celsius * 10. Clouded areas where no air temperature predictions could be made for are marked as no data. The size of each file is around 15 MB depending on the amount of no data values due to cloud conditions.

As secondary dataset, the daily AntAir data were aggregated to monthly means (Meyer et al., 2019b). No data values on individual days due to clouds were ignored in this procedure to provide a seamless dataset.
All data processing and modelling procedures are available as R scripts on a public Github repository:
www.github.com/HannaMeyer/AntAir.
R version 3.4 was used (R Core Team, 2018). Major packages included are the raster package (Hijmans, 2017) for handling and processing of raster data, caret (Kuhn, 2017) as a wrapper package for machine learning implementations and CAST (Meyer, 2018) for feature selection and spatial validation of the models.

*Competing interests.* The authors declare no competing interests

*Acknowledgements.* This research was supported by the Ross Sea Region Terrestrial Data Analysis research program, funded by the Ministry of Business and Innovation, New Zealand, with contract number CO9X1413. The authors appreciate the support of the Antarctic Meteorological Research Center for the data set of weather stations from University of Wisconsin-Madison (http://amrc.ssec.wisc.edu/). Data and information from the Italian weather stations of Antarctica were obtained from "MeteoClimatological Observatory at MZS and Victoria Land" of PNRA - http://www.climantartide.it. Further weather station data from the McMurdo Dry Valleys (Soil Climate Research Station Data Antarctica) were obtained from Soil Survey Staff, Natural Resources Conservation Service, United States Department of Agriculture.





Available online at http://www.nrcs.usda.gov/wps/

portal/nrcs/main/soils/survey/climate/. This study was conducted using the Palma II HPC system from the University of Münster. We further

acknowledge support from the Open Access Publication Fund of the University of Münster.





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
