# Peer review of "AntAir: satellite-derived 1 km daily Antarctic air temperatures since 2003"

_Earth System Science Data, 2019_

## Referee Comment (RC1) · Anonymous Referee #1 · 14 Feb 2020

This manuscript reported a dataset of daily and monthly averaged Antarctic air temperature at the resolution of 1km for the 2003-2017 period. This dataset is constructed using MODIS LST and automatic weather station (AWS) air temperature observations by means of machine learning algorithms, which are proved to be effective for reconstructing Antarctic air temperature in the authors' early work published in 2016. The uncertainty of the dataset is also estimated based on a cross validation method. This dataset is an important supplement for the remote sensing based on global dataset of air temperature by Hooker et al. (2018), also useful for estimating Antarctic climate change, and validating the simulation of regional climate models.

Generally, this manuscript is well organized and written, and figures are appropriate. It deserve publication in ESSD. However, before publication, the manuscript still requires

some following revisions.

Page 3 10-15: What about the quality of AWS observations? Can the data be trusted and is the quality the same at all of the observational stations? What kind of quality control is made for these observations? The availability of observed data is greatly inconsistent in time between weather stations. How large can the errors in reconstructed air temperature be due to this inconsistence?

Page 3 20-30: Also please add the description of the quality of MODIS LST. It is also necessary to discuss the suitability of MODIS LST for Antarctic air temperature estimate. See Wang et al. (2013) reported robust correlation between MODIS LST and air temperature over the Lambert glacier drain.

Page 5 5-10 I think that the authors should use the DEMs with higher accuracy, such as DEMs from Bamber et al. (2009) or Slater et al. (2018), rather than RAMP DEM

In my opinion, spatial distribution of errors of reconstructed air temperature at each AWS location should be shown.

To further estimate the accuracy, the constructed dataset should be compared with the previous temperature reconstruction for their overlapping period by Steig et al. (2009), O'Donnell et al. (2011), and Nicolas and Bromwich (2014).

References:

Bamber, J. L., J. L. Gomez-Dans, and J. A. Griggs. 2009. A new 1 km Digital Elevation Model of the Antarctic derived from combined satellite radar and laser Data – Part 1: Data and Methods. The Cryosphere, 3, 101-111. Nicolas and Bromwich (2014) New Reconstruction of Antarctic Near-Surface Temperatures: Multidecadal Trends and Reliability of Global Reanalyses, Journal of Climate, 27, 8070-8093

O'Donnell, R.,N. Lewis, S.McIntyre, and J. Condon, 2011: Improved methods for PCA-based reconstructions: Case study using the Steig et al. (2009) Antarctic temperature reconstruction. J. Climate, 24, 2099–2115

Slater, T., Shepherd, A., McMillan, M., Muir, A., Gilbert, L., Hogg, A. E., Konrad, H., and Parrinello, T. 2018. A new digital elevation model of Antarctica derived from CryoSat-2 altimetry. The Cryosphere, 12, 1551-1562.

Steig, E. J., D. P. Schneider, S. D. Rutherford, M. E. Mann, J. C. Comiso, and D. T. Shindell, 2009: Warming of the Antarctic ice-sheet surface since the 1957 International Geophysical Year. Nature, 457, 459–462.

Wang et al. (2013) A Comparison of MODIS LST Retrievals with in Situ Observations from AWS over the Lambert Glacier Basin, East Antarctica. International Journal of Geosciences, 2013, 4, 611-617

---

## Referee Comment (RC2) · Anonymous Referee #2 · 15 Feb 2020

The opening statement - lines 13 and 14 of page 1 - is thoroughly, demonstrably and emphatically false. Antarctica, essentially $14 \times 10^6$ km$^2$ of snow and ice (not counting winter sea ice), exists, for six months or more of each annual cycle, in a completely frozen state. Frozen = zero terrestrial ecology/biology/biodiversity. During the 'warm' season (SH summer), the minuscule areas ($< 5 \times 10^3$ km$^2$, 0.03% or less of the total area) of Antarctica not snow covered (for hydrological rather than temperature reasons), e.g. McMurdo dry valleys, support a tiny, desperate exotic (and fascinating) mini-ecosystem which has virtually no impact on hydrology or glaciology beyond its restricted boundaries. The biodiversity of Convey, which the authors like to cite, refers to sub-Antarctic islands wherever BAS operates long-term research bases; note that Pete's very good work rarely if ever refers to British Halley Station on the continent (adjoined ice shelf) itself. Biodiversity issues for Antarctica, including invasive species, habitat (sea ice) modification or reduction, competition with human predators (e.g. for krill) occur almost exclusively in the marine realm. Likewise for proposed protection areas. No liquid-water hydrology exists at the surface of Antarctica. Extensive glacier mass balances and motions have little to no dependence on surface air temperature. Snow surface halogen chemistry, particularly within regions exposed to wind-blown sea salt aerosols, does show temperature dependence, on reaction rates and - to less extent - on products, but the authors seem blind to that entire field. Their deep ignorance of Antarctica, even if they had a valid surface air temperature product (which they do not), disqualifies their entire concept from the start. One wishes they might have read some early cross-ice explorers (e.g Behrendt) or explored IPY blogs from Norwegian or US transects. Do they even know about the snow road? Have they ever heard of crevices, sastrugi, nunataks? Do they know the Mawson story? They demonstrate no competence whatsoever.

Do the authors not understand 24-hour polar night alternating with polar constant daylight? They provide a daytime nighttime data extraction routine (page 3 line 27) which, one can scarcely believe, apparently ignores the entire issue of seasonal light levels (complete light, complete dark). Later (on page 5, paragraph starting at line 5) they describe use of solar angles to calculate hillshading as one of their predictor variables but they give no indication that they understand Antarctica; the description sounds more relevant to mid-latitude Germany.

One appreciates mention and use of the RadarSAT DEM, but even 200 m resolution (which they interpolate to 1000 m) misses most relevant surface texture. Higher-resolution airborne radar surveys over large areas of the ice sheet show flat smooth areas of various extents (over basal lakes) amidst much rougher ridged and fractured ice, often (evidently) with substantial temporal evolution. Again, they apparently have no idea. Their predictors have no relevance.

Their primary tool, MODIS LST, has demonstrated and much-argued weaknesses over snow and ice, both for cloud masks and surface temperature extractions. One might have hoped that Meyer et al (2016, the predecessor to this work and again much cited by these authors) might have addressed if not offered new resolution to some of those known issues but that paper blithely accepts MODIS products (citing primarily mid-latitude terrestrial examples) as de facto valid despite a large, vociferous and continuing debate about applicability, suitability and errors over snow and ice. Until or unless these authors demonstrate and document new algorithms or techniques to improve performance of MODIS products over snow and ice they and we must regard this particular application as un-proven at best. A large literature, none of if cited here, debates these troublesome issues of single or multiple sensors and their individual or combined effectiveness at retrieving surface air temperatures over snow and ice. Again, the authors demonstrate no competence whatsoever in the use of MODIS LST.

After casual application of four different machine learning techniques, the authors in the end **rely on visual inspection!!!** Their complete inability, despite multiple runs of multiple software tools, extensive spatial and leave-out cross-validation, to rely on any single outcome despite extensive statistical evaluations disqualifies the entire effort. This potential user might have asked for fine-scale validation in areas of (relatively dense) met measurements or perhaps RMSE sorted by elevation, but why bother? They literally have nothing valid to show.

To report absolute and RMS errors of 5K seems absurd. Who do they think might use such imprecise unreliable data? From any of several authors (try anything from Scambos, for example) these authors should know concern about long-term climate induced trends of 0.3 to 0.4 $^o$C per decade over higher elevations of East Antarctica. Here they can't provide better than 5K? Over 15 years (assuming their time period of 2003 to 2018 (but apparently 2003 to 2016 according to page 11 line 6)), we might expect temperature change of perhaps 0.6 $^o$C? Even by yearly averaging (1.73 $^o$C, page 11 line 5), they fail to come close to necessary precision. They refer (e.g. page 8 line 31) to "RF being superior in the temporal prediction" but they fail entirely to demonstrate necessary temporal skill. Again, once senses that they fundamentally do not understand the system they attempt to model. What, by the way, do bold values in Tables 1 or 2 indicate? Some kind of statistical certainty of statistical summaries? And what do the axis units in Figure 4 indicate? One gets the strong sense that we have gone substantially backward in precision, accuracy and reliability with this product.

I find a several other conceptual errors (which only reinforce a sense that these authors - despite apparent mathematical skills - have not the faintest idea of the Antarctic environment) and several of language, but why bother? I sincerely regret that I took on the task to serve as reviewer. I can well understand that many others declined to review. Perhaps these authors will receive a different more-favorable review. This reviewer emphatically recommends complete rejection: ESSD will damage its admirable reputation by publishing such nonsense

---

## Editor Comment (EC1) · David Carlson (Editor) · 17 Apr 2020

Despite a title that implies a product of Antarctic-wide impact, coupled with an expansive "climatology, hydrology or ecology" first sentence, these authors - by their own words - motivate and justify their work solely by its relevance to ice-free areas: "we're mainly talking of ice-free areas when it comes to terrestrial biodiversity". Although they claim to respond to reviewer 2 (R2) "point-by-point", I found nothing responding to the "0.03%" affected land area claim by R2. Do authors have information to refute that number? Can they show quantitatively that the relevance of their work for "biodiversity" extends to larger areas of the Antarctic continent? Can they cite Antarctic or polar or even temperate biological or biogeochemical pathways or processes for which a temperature resolution of 5K would prove sufficient? The single reference they

cite (Wauchope et al. 2019) deals exclusively with protected areas, shows an areal extent image (Figure 1 in Wauchope) that appears to prove R2's point about vast areas of frozen Antartica, and provides emphasis of narrow applicability of this work: "Terrestrial biodiversity in the Antarctic is predominantly restricted to areas that are permanently ice-freeâ̆Tcurrently estimated at somewhere between 0.2 and 0.5% of the Antarctic continent." Hogg & Wall (a citation that appears later in their response to R2) represents an overview document for a special issue - Hogg & Wall introduce work of others but do not present their own evidence. These authors appear not to contend R2's point about most of Convey's work and sub-Antarctic islands?

Hogg & Wall repeat the point about temperature trends: "roughly 1°C" over some unspecified (50 years?) time period. These authors dodge or avoid the question of relevance of RMSE of 5K. Instead they admit: "Indeed, we have higher errors than in other studies related to air temperature" and "the validation statistics are less impressive than in a lot of studies". They follow with a list of machine learning citations none of which mention Antarctica and none of which an Antarctic researcher will ever read. Their RMSE vs elevation graphic, included in response to R2, paints an even worse picture. Compared with Figure 1 of Wauchope, these authors show worse RMSE (around 8K) in coastal and perimeter regions where improved air temperatures might prove useful to biodiversity while their lowest RMSE (4 to 5K) occurs in 'bio-deserts' high on the ice sheet? (Also, according to their own Figure 1, worse RMSE where they have higher abundance of situ validation data? Does that not give them pause?)

About the use of LandSat LST over snow- or ice-covered regions - a concern raised by both reviewers - they conclude only "This is an ongoing challenge and further research effort on this will certainly improve the presented AntAir dataset in the future."

This product involves application of four machine learning algorithms - Random Forests, Generalized Boosted Regression Models, Neural Networks, and Partial Least Squares Regression - chosen "because they are either frequently applied algorithms or have shown good performance in the pilot study of Meyer et al. (2016)". Meyer et

al looked only at GBM (with a preliminary RMSE of 11K!), so how can they cite that as qualifying distinction? If this does not represent a "casual" (in words of R2) assembly of algorithms, these authors certainly have not given readers or data users sufficient information to understand why these four? Do none of the four algorithms have prior effective published use for air temperatures in polar regions? I found the comment by R2 about 'visual inspection' curious but also relevant. Would a different visual inspector have have arrived at a different conclusion? How would one certify or replicate 'visual inspection' in a validation sense? These authors reference a satellite product that requires "further research" with outcomes that (again, in their words), refer, sensu stricto, to "ice-free areas" with "higher errors than in other studies" and "validation statistics are less impressive than in a lot of studies".

ESSD exists to reward data providers for sharing useful high-quality data with a large community of users. I fail to see how these authors have provided sufficient justification or certification of quality.

---

## Author Comment (AC1) · 17 Apr 2020

**Reply to Anonymous Referee #1**

This manuscript reported a dataset of daily and monthly averaged Antarctic air temperature at the resolution of 1km for the 2003-2017 period. This dataset is constructed using MODIS LST and automatic weather station (AWS) air temperature observations by means of machine learning algorithms, which are proved to be effective for reconstructing Antarctic air temperature in the authors' early work published in 2016. The uncertainty of the dataset is also estimated based on a cross validation method. This dataset is an important supplement for the remote sensing based on global dataset of air temperature by Hooker et al. (2018), also useful for estimating Antarctic climate change, and validating the simulation of regional climate models. Generally, this manuscript is well organized and written, and figures are appropriate. It deserve publication in ESSD. However, before publication, the manuscript still requires some following revisions.

Thank you for your positive feedback. We hope that our response will meet your expectation.

Page 3 10-15: What about the quality of AWS observations? Can the data be trusted and is the quality the same at all of the observational stations? What kind of quality control is made for these observations? The availability of observed data is greatly inconsistent in time between weather stations. How large can the errors in reconstructed air temperature be due to this inconsistence?

We used the best available data for this region but certainly maintenance is challenging in such a remote area. However, as of the inconsistencies in time we are not expecting any problems because we assume that relationships between LST, further predictors, and air temperature don't change. So it is not required that we have complete observations for each AWS, as long as the gradients in the predictors are well covered in the data. We will add a section to the manuscript on this topic:

*"It is extremely difficult to maintain Antarctic remote stations like these. With the data used here we rely on the utmost professional ability of the programs within the very constraining international level logistics and costs of Antarctic operations. There has been no apparent inter-agency harmonization of sensor types we are aware of and the data quality check of the data used here was limited to the effort of the individual providers. The data are not consistent in terms of their time series. However, due to the machine learning approach that is used for the spatio-temporal model of air temperature, it is not required here that a full time series at a respective location is acquired: The task of the algorithm is to learn the relationships between satellite-based LST as well as environmental properties and air temperature. Since long time series are used, we are confident that the general gradients in the data are covered"*

Page 3 20-30: Also please add the description of the quality of MODIS LST. It is also necessary to discuss the suitability of MODIS LST for Antarctic air temperature estimate. See Wang et al. (2013) reported robust correlation between MODIS LST and air temperature over the Lambert glacier drain.

We will add the following description on the MODIS LST quality to the manuscript.

*"The data are cloud-masked using the MODIS Cloud Mask algorithm (Ackerman et al., 1998) that applies typical thresholds in the visible and infrared channels. Though the MODIS LST product is cloud masked, the "white on white" and "cold on cold" effect is a challenge for cloud detection in Antarctica (Allen et al., 1990). This holds especially true for cirrus clouds that could in parts not reliably be detected in the used LST product. This is an ongoing challenge and further research effort on this will certainly improve the presented AntAir dataset in the future.*
*The MODIS LST data are reported with a quality of better than 1°C in the range from -10 to 50°C (Wan et al., 2004). However this did not involve an extensive validation for Antarctica. For the*

*antarctic McMurdo Dry Valleys, Wan (2014) reported a mean error of 1K. Note that a general bias is not problematic for this study due to the applied machine learning based regression approach, but that robust relationships is relevant. Here, previous studies have indicated robust correlation between MODIS LST and measured air temperature, e.g. Wang et al., 2013 over the Lambert glacier drain or Li et al. (2016) between measured snow surface temperature and MODIS LST."*

Page 5 5-10 I think that the authors should use the DEMs with higher accuracy, such as DEMs from Bamber et al. (2009) or Slater et al. (2018), rather than RAMP DEM

The RAMP DEM might have an accuracy not sufficient for e.g. glacier drainage basin delineation for mass balance analyses (Cook et al., 2012), however has been indicated to be suitable as a surface topography dataset (Cook et al., 2012), which is what we need it for.

Please note that not all the characteristics of the DEM were used as predictor (see results after variable selection), so the only terrain related relevant information is daily maximum hillshade. We compared the results for daily maximum hillshade for the DEM of Bamber et al. with hillshade derived from RAMP and found that the differences are small ($R^2$ for the entire study area = 0.98 and with focus on the DryValleys where hillshading is most variable= 0.84). Therefore we are convinced that no change in the results must be expected.

We will justify our choice of the DEM in the manuscript: *"The Radarsat Antarctic Mapping Project (RAMP) digital elevation model (DEM) (Liu et al., 2015), version 2, was used, which has been indicated to be suitable as a surface topography dataset (Cook et al., 2012)"*

Cook, A. J., Murray, T., Luckman, A., Vaughan, D. G., and Barrand, N. E.: A new 100-m Digital Elevation Model of the Antarctic Peninsula derived from ASTER Global DEM: methods and accuracy assessment, Earth Syst. Sci. Data, 4, 129–142, https://doi.org/10.5194/essd-4-129-2012, 2012.

In my opinion, spatial distribution of errors of reconstructed air temperature at each AWS location should be shown.

Agreed, this will add valuable information. We will include the following map to the manuscript, that shows the error/performance for each station.

[Figure]

To further estimate the accuracy, the constructed dataset should be compared with the previous temperature reconstruction for their overlapping period by Steig et al. (2009),

O'Donnell et al. (2011), and Nicolas and Bromwich (2014).

We appreciate this comment and we're sorry that we cannot fulfill the request for the following reasons: A comparison based on statistical validation metrics as reported in the publications is not possible because of different data, validation strategy etc being used. So only a direct comparison using the data is an option. Unfortunately, for all three data sets that you mentioned only anomalies can be accessed ([http://polarmet.osu.edu/datasets/Antarctic_recon/](http://polarmet.osu.edu/datasets/Antarctic_recon/), [http://faculty.washington.edu/steig/nature09data/data/](http://faculty.washington.edu/steig/nature09data/data/)), which cannot be compared to our absolute temperature values. If you're aware of e.g. monthly temperature data available for the mentioned papers, please let us know and we will be happy to perform a direct comparison.

---

## Author Comment (AC2) · 17 Apr 2020

**Reply to Anonymous Referee #2**

Authors demonstrate no competence for the topic of Antarctica. Product has no value.

We are well aware of the fact that this response will not change the reviewers opinion on our work. However, we want to act professionally, and will endevour to reply to comments provided that these were articulated appropriately.
We strongly refute the claim that we "demonstrate no competence for the topic of Antarctica": the team builds on a wealth of experience on Antarctic environments for the last 15 years, and has a combined field experience of approx. 15 seasons.

The opening statement - lines 13 and 14 of page 1 - is thoroughly, demonstrably and emphatically false. Antarctica, essentially 14 x 106 km2 of snow and ice (not counting winter sea ice), exists, for six months or more of each annual cycle, in a completely frozen state. Frozen = zero terrestrial ecology/biology/biodiversity. During the 'warm' season (SH summer), the minuscule areas (< 5 x 103 km2, 0.03% or less of the total area) of Antarctica not snow covered (for hydrological rather than temperature reasons), e.g. McMurdo dry valleys, support a tiny, desperate exotic (and fascinating) mini-ecosystem which has virtually no impact on hydrology or glaciology beyond its restricted boundaries. The biodiversity of Convey, which the authors like to cite, refers to sub-Antarctic islands wherever BAS operates long-term research bases; note that Pete's very good work rarely if ever refers to British Halley Station on the continent (adjoined ice shelf) itself. Biodiversity issues for Antarctica, including invasive species, habitat (sea ice) modification or reduction, competition with human predators (e.g. for krill) occur almost exclusively in the marine realm. Likewise for proposed protection areas. No liquid-water hydrology exists at the surface of Antarctica. Extensive glacier mass balances and motions have little to no dependence on surface air temperature. Snow surface halogen chemistry, particularly within regions exposed to wind-blown sea salt aerosols, does show temperature dependence, on reaction rates and - to less extent - on products, but the authors seem blind to that entire field.

The following paper demonstrates clearly that biodiversity is, by far, not limited to the McMurdo Dry Valleys.

- Wauchope, H., Shaw, J.D. & Terauds, A. A snapshot of biodiversity protection in Antarctica. *Nat Commun* **10,** 946 (2019). https://doi.org/10.1038/s41467-019-08915-6

There is also a wealth of literature (as cited below) supporting the importance of near-surface air temperature as a driver of *terrestrial biodiversity, and hydrological and glaciological processes.* For a revision, we will support our statement by further literature and we will slightly rephrase it to make clear that we're mainly talking of ice-free areas when it comes to terrestrial biodiversity:

*"Near-surface air temperature in Antarctica is an important driver of terrestrial biodiversity in the ice-free areas(Convey&Smith, 2006; Convey, 2010; Hogg et al., 2011) and is decisive for hydrological (Herbei et al., 2016) and glaciological processes (Cook et al., 2005). "*

- Hogg, I.D., Wall, D.H. Global change and Antarctic terrestrial biodiversity. *Polar Biol* **34,** 1625 (2011). https://doi.org/10.1007/s00300-011-1108-9
- Convey, P., Smith, R.I.L. Responses of Terrestrial Antarctic Ecosystems to Climate Change. *Plant Ecol* **182,** 1–10 (2006). https://doi.org/10.1007/s11258-005-9022-2

Do the authors not understand 24-hour polar night alternating with polar constant daylight? They provide a daytime nighttime data extraction routine (page 3 line 27) which, one can scarcely believe, apparently ignores the entire issue of seasonal light levels (complete light, complete dark).

We are fully aware of 24-hour polar night alternating with polar constant daylight. MODIS is a global dataset that provides 2 acquisitions a day, one called "day-time" the other being named "night-time". Obviously in Antarctica the name of those products can be a little misleading.

Later (on page 5, paragraph starting at line 5) they describe use of solar angles to calculate hillshading as one of their predictor variables but they give no indication that they understand Antarctica; the description sounds more relevant to mid-latitude Germany.

Certainly there are hillshading effects in Antacrtica as well which affect the temperature. This is certainly nothing that applies to mid-latitude Germany only. Complete darkness (same as complete sunlight) has been considered since the hillshading is used here as a temporally dynamic variable as extensively described in the manuscript.

One appreciates mention and use of the RadarSAT DEM, but even 200 m resolution (which they interpolate to 1000 m) misses most relevant surface texture. Higher-resolution airborne radar surveys over large areas of the ice sheet show flat smooth areas of various extents (over basal lakes) amidst much rougher ridged and fractured ice, often (evidently) with substantial temporal evolution. Again, they apparently have no idea. Their predictors have no relevance.

We are working on 1 km resolution in agreement with the MODIS LST data. Though higher resolution DEMs exist, LST can, with this temporal resolution, only be provided at 1 km spatial resolution. We agree that 1 km misses a lot of the high resolution features, however, 1 km is still a huge improvement upon the existing temperature products available at continental scale.

Also the first reviewer was asking about the accuracy of the DEM and was suggesting using the DEMs from Bamber et al. (2009) or Slater et al. (2018), rather than RAMP DEM.

Although the RAMP DEM might have an accuracy not sufficient for e.g. glacier drainage basin delineation for mass balance analyses (Cook et al., 2012), however has been indicated to be suitable as a surface topography dataset (Cook et al., 2012), which is what we need it for. Please note that not all the characteristics of the DEM were used as predictor (see results after variable selection), so the only terrain related relevant information is daily maximum hillshade. We compared the results for daily maximum hillshade for the DEM of Bamber et al. with hillshade derived from RAMP and found that the differences are small ($R^2$ for the entire study area = 0.98 and with focus on the DryValleys where hillshading is most variable= 0.84). Therefore we are convinced that no change in the results must be expected.

We will justify our choice of the DEM in the manuscript: "*The Radarsat Antarctic Mapping Project (RAMP) digital elevation model (DEM) (Liu et al., 2015), version 2, was used, which has been indicated to be suitable as a surface topography dataset (Cook et al., 2012)*"

Their primary tool, MODIS LST, has demonstrated and much-argued weaknesses over snow and ice, both for cloud masks and surface temperature extractions. One might have hoped that Meyer et al (2016, the predecessor to this work and again much cited by these authors) might have addressed if not offered new resolution to some of those known issues but that paper

blithely accepts MODIS products (citing primarily mid-latitude terrestrial examples) as de facto valid despite a large, vociferous and continuing debate about applicability, suitability and errors over snow and ice. Until or unless these authors demonstrate and document new algorithms or techniques to improve performance of MODIS products over snow and ice they and we must regard this particular application as un-proven at best. A large literature, none of if cited here, debates these troublesome issues of single or multiple sensors and their individual or combined effectiveness at retrieving surface air temperatures over snow and ice. Again, the authors demonstrate no competence whatsoever in the use of MODIS LST.

*Also following the recommendation of the first reviewer, we included a more detailed information on the MODIS LST product.*

*"The data are cloud-masked using the MODIS Cloud Mask algorithm (Ackerman et al., 1998) that applies typical thresholds in the visible and infrared channels. Though the MODIS LST product is cloud masked, the "white on white" and "cold on cold" effect is a challenge for cloud detection in Antarctica (Allen et al., 1990). This holds especially true for cirrus clouds that could in parts not reliably be detected in the used LST product. This is an ongoing challenge and further research effort on this will certainly improve the presented AntAir dataset in the future.*
*The MODIS LST data are reported with a quality of better than 1°C in the range from -10 to 50°C (Wan et al., 2004). However this did not involve an extensive validation for Antarctica. For the antarctic McMurdo Dry Valleys, Wan (2014) reported a mean error of 1K. Note that a general bias is not problematic for this study due to the applied machine learning based regression approach, but that robust relationships is relevant. Here, previous studies have indicated robust correlation between MODIS LST and measured air temperature, e.g. Wang et al., 2013 over the Lambert glacier drain or Li et al. (2016) between measured snow surface temperature and MODIS LST."*

After casual application of four different machine learning techniques, the authors in the end rely on visual inspection!!! Their complete inability, despite multiple runs of multiple software tools, extensive spatial and leave-out cross-validation, to rely on any single outcome despite extensive statistical evaluations disqualifies the entire effort.

*We disagree with the reviewer's view on the model validation strategy, and found the wording of a "casual application" surprising, as the procedure of extensively tuned and validated models go far beyond most commonly seen validation strategies.*

*Moreover, in this paper we are not "relying" on visual inspection, rather, but we acknowledge visual inspection as an additional validation step, in addition to the outlined a) spatial and b) temporal validation by independent data. It is widely documented that blindly looking at statistical values is very dangerous when analysing what flexible algorithms actually learn (this is a large topic in machine learning, see e.g. Lapuschkin et al., 2020 or Patrick Schramowski et al., 2020). We ensured that the algorithm is learning meaningful relationships (meaningful in terms of spatial and temporal prediction for new unseen data) by spatial variable selection, spatial and temporal statistical validation, and a final expert inspection of the model predictions by visual analysis. It has been shown by many other studies that it is not sufficient to look at statistical values, and that expert knowledge should not be left out in a validation procedure.*

- *Lapuschkin, S., Wäldchen, S., Binder, A. et al. Unmasking Clever Hans predictors and assessing what machines really learn. Nat Commun 10, 1096 (2019). https://doi.org/10.1038/s41467-019-08987-4*
- *Schramowski et al. (2020): Right for the Wrong Scientific Reasons: Revising Deep Networks by Interacting with their Explanations. arXiv:2001.05371*

This potential user might have asked for fine-scale validation in areas of (relatively dense) met measurements or perhaps RMSE sorted by elevation, but why bother?

Following the constructive suggestions of the first reviewer, we included a figure showing the RMSE/R² per station. We can also provide a figure for a revised manuscript, showing the RMSE sorted by elevation, although, no significant patterns are obvious.

[Figure]

They literally have nothing valid to show. To report absolute and RMS errors of 5K seems absurd. Who do they think might use such imprecise unreliable data? From any of several authors (try anything from Scambos, for example) these authors should know concern about long-term climate induced trends of 0.3 to 0.4 oC per decade over higher elevations of East Antarctica. Here they can't provide better than 5K? Over 15 years (assuming their time period of 2003 to 2018 (but apparently 2003 to 2016 according to page 11 line 6)), we might expect temperature change of perhaps 0.6 oC? Even by yearly averaging (1.73 oC, page 11 line 5), they fail to come close to necessary precision. They refer (e.g. page 8 line 31) to "RF being superior in the temporal prediction" but they fail entirely to demonstrate necessary temporal skill. Again, once senses that they fundamentally do not understand the system they attempt to model.

Indeed, we have higher errors than in other studies related to air temperature estimation from satellite, but we do not believe this is due to "going backward in precision", but rather to the very strict validation strategy that we used here. We validate our model using independent spatial

locations and years different from what has been used for model training, and argue that while the validation statistics are less impressive than in a lot of studies, it is also a lot more realistic. We refer to Roberts et al. (2017), Meyer et al. (2018), Valavi et al. (2018), Pohjankukka et al. (2017) to support our claim.

- Roberts, D. R., Bahn, V., Ciuti, S., Boyce, M. S., Elith, J., Guillera-Arroita, G., Hauenstein, S., Lahoz-Monfort, J. J., Schröder, B., Thuiller, W., Warton, D. I., Wintle, B. A., Hartig, F. & Dormann, C. F. (2017), 'Cross-validation strategies for data with temporal, spatial, hierarchical, or phylogenetic structure', Ecography.
- Meyer, H., Reudenbach, C., Hengl, T., Katurji, M. & Nauss, T. (2018), 'Improving performance of spatio-temporal machine learning models using forward feature selection and target-oriented validation', Environmental Modelling & Software 101, 1– 9.
- Pohjankukka, J., Pahikkala, T., Nevalainen, P. & Heikkonen, J. (2017), 'Estimating the prediction performance of spatial models via spatial k-fold cross validation', International Journal of Geographical Information Science 31(10), 2001–2019.
- Valavi, R., Elith, J., Lahoz-Monfort, J. J. & Guillera-Arroita, G. (2018), 'blockcv: an r package for generating spatially or environmentally separated folds for k-fold cross-validation of species distribution models', Methods in Ecology and Evolution.

What, by the way, do bold values in Tables 1 or 2 indicate? Some kind of statistical certainty of statistical summaries? And what do the axis units in Figure 4 indicate?

The caption of the tables 1 and 2 point on that: "The best performances are highlighted". We will include the more detailed information that they are "highlighted in bold" for a revised version. We will also add the coordinate reference system to the units of Figure 4.

One gets the strong sense that we have gone substantially backward in precision, accuracy and reliability with this product.

See our comment above. Also, there is currently no spatio-temporal dataset we are aware of, that has a better performance in this spatio-temporal resolution.

---

## Editor Comment (EC2) · David Carlson (Editor) · 19 Apr 2020

I apologize for a small error in my comment. I wrote 'Landsat' when I should have written 'MODIS'. Weak excuse, but at the time I wrote those comments I also searched for most-recent cloud-free LandSat scene for my region. Name error does not change the concern, that MODIS LST remains a challenging product when calculated/estimated over snow or ice.

---

## Author Comment (AC3) · 25 Apr 2020

The presented dataset is able to reflect spatio-temporal patterns of air temperature in a high spatial and temporal resolution. We accept the concern that the accuracy could be further improved, and respect the decision to reject the dataset for this reason despite receiving a very supportive review from R1. However, we fundamentally disagree with the majority of the other points that were raised and it is important for us to express our opinion on a few points:

Apparently, a major concern of both R2 and the editor, is on the fact that biodiversity patterns are one (of several) motivations to develop this dataset ("*I found nothing responding to the "0.03%" affected land area claim by R2. Do authors have information to refute that number?*"). We're not refuting the percentage, and don't see the need to do so for our air temperature dataset. We explicitly state that the motivation of our dataset is intended for application in the context of several disciplines ("*Air temperature is an important baseline parameter for terrestrial Antarctica in the context of patterns and processes in climatology, hydrology or ecology.*"). Just because biodiversity is only limited to the ice-free areas does not mean that our air temperature product is limited to the same. The product itself, the figures given in the manuscript, as well as the text make clear that the dataset provides air temperature for entire Antarctica .

Both, the choice of the algorithm, as well as the validation strategy are described as casual by the editor and R2 respectively. We strongly disagree, as we outlined in our response to R2. Also, just as a minor remark, the point "*Meyer et al looked only at GBM*", which led the editor to the impression that the selection of algorithms is rather poorly justified, is not correctly cited in the review: already in the abstract of this paper, as well as in several other places, the cited paper emphasizes that "*the performance of a simple linear regression model to predict Tair from LST was compared to the performance of three machine learning algorithms: Random Forest (RF), generalized boosted regression models (GBM) and Cubist.*"

*For the Editor's comment on: "Also, according to their own Figure 1, worse RMSE where they have higher abundance of situ validation data? Does that not give them pause".* This is not the case here. In the area with high density, some of the stations have rather high errors, but at the same time there are stations for which the error is very low (see additional figure below that focus on the area of the MDV where we have the highest density). This is very positive as we show that no general bias towards the areas with high density occurs.

[Figure]

Finally, the point that we very fundamentally disagree with, is that current trends and discussions on spatial predictive mapping developed in a context that might not be related to Antarctica is not of relevance here (*"They follow with a list of machine learning citations none of which mention Antarctica and none of which an Antarctic researcher will ever read."*). We are convinced that interdisciplinary collaboration is needed in the field of environmental monitoring and believe that this is the condition to make progress towards improved monitoring of environmental variables, both in Antarctica as well as elsewhere.

Finally, and despite our disagreement on some of the raised concerns, we would like to thank the reviewers and the editor for their work on our manuscript.

---

## Editor Comment (EC3) · David Carlson (Editor) · 27 Apr 2020

With sufficient comments and complaints now in the record, authors should withdraw their manuscript.